# Association of Calcitriol Supplementation with Reduced COVID-19 Mortality in Patients with Chronic Kidney Disease: A Population-Based Study

**DOI:** 10.3390/biomedicines9050509

**Published:** 2021-05-05

**Authors:** Joaquim Oristrell, Joan Carles Oliva, Isaac Subirana, Enrique Casado, Didier Domínguez, Andrea Toloba, Patricia Aguilera, Joan Esplugues, Pilar Fafián, Maria Grau

**Affiliations:** 1Internal Medicine Service, Parc Taulí Health Corporation Consortium, 08208 Sabadell, Catalonia, Spain; paguilera@tauli.cat (P.A.); jesplugues@tauli.cat (J.E.); pfafian@tauli.cat (P.F.); 2Parc Taulí Research and Innovation Institute (I3PT), 08208 Sabadell, Catalonia, Spain; jcoliva@tauli.cat; 3Department of Medicine, Autonomous University of Barcelona, 08193 Barcelona, Catalonia, Spain; 4Consortium for Biomedical Research in Epidemiology and Public Health (CIBERESP), 08003 Barcelona, Catalonia, Spain; isubirana@imim.es (I.S.); atoloba@imim.es (A.T.); 5Hospital del Mar Medical Research Institute (IMIM), 08003 Barcelona, Catalonia, Spain; 6Rheumatology Service, Parc Taulí Health Corporation Consortium, 08208 Sabadell, Catalonia, Spain; ecasado@tauli.cat; 7Agency for Health Quality and Assessment of Catalonia (AQUAS), 08005 Barcelona, Catalonia, Spain; didier.dominguez@gencat.cat; 8Serra Hunter Fellow, Department of Medicine, University of Barcelona, 08036 Barcelona, Catalonia, Spain

**Keywords:** calcitriol, vitamin D, COVID-19, SARS-CoV2 infection, chronic kidney disease

## Abstract

Treatment with calcitriol, the hormonal form of vitamin D, has shown beneficial effects in experimental models of acute lung injury. In this study, we aimed to analyze the associations between calcitriol supplementation and the risk of SARS-CoV2 infection or COVID-19 mortality. Individuals ≥18 years old living in Catalonia and supplemented with calcitriol from April 2019 to February 2020 were compared with propensity score matched controls. Outcome variables were SARS-CoV2 infection, severe COVID-19 and COVID-19 mortality. Associations between calcitriol supplementation and outcome variables were analyzed using multivariable Cox proportional regression. A total of 8076 patients were identified as being on calcitriol treatment. Advanced chronic kidney disease and hypoparathyroidism were the most frequent reasons for calcitriol supplementation in our population. Calcitriol use was associated with reduced risk of SARS-CoV2 infection (HR 0.78 [CI 95% 0.64–0.94], *p* = 0.010), reduced risk of severe COVID-19 and reduced COVID-19 mortality (HR 0.57 (CI 95% 0.41–0.80), *p* = 0.001) in patients with advanced chronic kidney disease. In addition, an inverse association between mean daily calcitriol dose and COVID-19 severity or mortality was observed in treated patients, independently of renal function. Our findings point out that patients with advanced chronic kidney disease could benefit from calcitriol supplementation during the COVID-19 pandemic.

## 1. Introduction

Infection with the new coronavirus SARS-CoV2 is characterized by an important clinical variability, ranging from completely asymptomatic cases to patients who develop a systemic disease (COVID-19) with severe lung involvement and high mortality. This clinical heterogeneity and the fact that the disease more severely affects older individuals with associated comorbidities [1] suggests that there are host-related factors that are of upmost relevance in the pathogenesis and prognosis of COVID-19.

ACE2 is the major cellular receptor for SARS-CoV2 [2] and SARS-CoV. ACE2 is a carboxypeptidase that catalyzes the synthesis of the protective, vasodilator and anti-inflammatory peptide Angiotensin (1–7) [3,4]. In addition, ACE2 has proteolytic effects on the proinflammatory mediators lys-des-Arg-bradykinin and des-Arg-bradykinin [5]. Severe lung injury induced by SARS-CoV has been associated with reduced lung expression of ACE2 [6], and low ACE2 expression has also been found in upper airways of patients infected by SARS-CoV2 [7]. 

Calcitriol (1,25-dihydroxycholecalciferol, the hormonal form of vitamin D) can protect against infections via an increase in the production of LL-37, β-defensin2 and nitric oxide (NO) in respiratory epithelia [8]. In addition, calcitriol has been shown to reduce the incidence of adult respiratory distress syndrome in experimental models of lipopolysaccharide-induced acute lung injury [9,10,11]. These beneficial effects were associated with calcitriol induction of pneumocyte II ACE2 expression [11,12]. For all these reasons, we hypothesized that calcitriol supplementation could protect against COVID-19. 

There are several ongoing clinical trials that will shed light in the future on the effects of vitamin D supplementation on COVID-19 outcomes [13]. At present, there have been a few small-sized studies that have analyzed the effects of cholecalciferol or calcifediol supplementation on COVID-19 outcomes, but to the best of our knowledge, none have studied the effects of calcitriol.

In this large, population-based, observational study we aimed to analyze the associations between calcitriol supplementation and the risks of infection or death from COVID-19. Since calcitriol is mainly prescribed to patients with advanced chronic kidney disease, we also made a subanalysis in this subgroup of patients. 

## 2. Materials and Methods

### 2.1. Population Included and Study Design

We analyzed all individuals ≥18 years old insured by the Catalan public health System that were alive on 25 February 2020, the date of the first positive PCR for SARS-CoV2 (*n* = 6,348,094). 

In this population we identified all patients being supplemented with calcitriol from 1 April 2019 to 28 February 2020 (*n* = 8076) and the patients that had not been supplemented with any vitamin D compound (5,848,776). Patients without an available serum creatinine determination were excluded for further analysis. After propensity score matching (see below), 6252 subjects on calcitriol and 12,504 matched controls were selected for study. In addition, all patients being supplemented with calcitriol between November 2019 and February 2020 (*n* = 5885) were selected to analyze the associations between calcitriol dosing and COVID-19 outcomes. 

### 2.2. Data Sources

Given Catalonia’s universal health and medication coverage, we were able to utilize electronic databases to examine the association of calcitriol use with COVID-19 outcomes in a real-world setting. We used anonymized data provided by the Agency for Health Quality and Assessment of Catalonia (AQUAS) within the framework of the Data Analytics Program for Health Research and Innovation (PADRIS). PADRIS databases include information on demographics (age and sex), diagnoses, laboratory data, drugs supplied by pharmacies, Primary Care physician diagnoses, laboratory results and diagnoses, procedures and outcomes of medical admissions in the public hospitals in Catalonia. This project was approved in a public call for grants for using PADRIS databases in research projects on COVID-19. 

### 2.3. Identification of Patients on Calcitriol Supplementation

Patients who had been supplied calcitriol by pharmacies (Anatomical Therapeutic Chemical Classification System group A11CC04) from 1 April 2019 to 28 February 2020 were analyzed. The sum of the Defined Daily Doses (DDD) of calcitriol supplied from 1 November 2019 to 28 February 2020 was identified and transformed into micrograms, and the mean daily calcitriol dose received per patient, in micrograms, was calculated.

### 2.4. Identification of Control Subjects through Propensity Score Matching

We performed a propensity score matching to build the control group using the ‘Matching’ package in R [14]. Since chronic kidney disease (CKD) is a strong predictor of worse prognosis in COVID-19 [15] and calcitriol is often prescribed in patients diagnosed with CKD, subjects without an available serum creatinine determination performed between 1 October 2018 and 28 February 2020 were excluded for matching. First, we used multivariate logistic regression to model receiving or not receiving calcitriol as a function of the following covariates: sex, age, 14 comorbidities identified from the International Classification of Diseases (ICD-10) diagnostic codes issued by family physicians (Appendix A), estimated glomerular filtration rate (eGFR), history of cigarette smoking, nursing home residence and use of seven classes of drugs that could potentially affect the prognosis (Appendix A). Estimated glomerular filtration rate was obtained from serum levels of creatinine, sex and age according to the Chronic Kidney Disease Epidemiology Collaboration (CKD-EPI) equation [16]. 

Propensity scores were matched using the nearest-neighbor matching method without replacement at a 1:2 ratio of treated subjects and controls. A caliper of 0.2 of the standard deviation of the propensity score logit was established as the maximum tolerated difference between matched patients. To examine the balance of each covariate between the treatment and the control group, the standardized mean difference was calculated before and after matching using the Tableone package in R [17]. We considered the groups well balanced if the standardized mean difference was <0.10 for each covariate. 

### 2.5. Outcome Variables

We analyzed the occurrence of SARS-CoV2 infection, COVID-19 hospitalization, intensive care admission, the procedures during hospitalization and mortality during the first wave of the pandemic. Three main outcome variables were defined, with different timings due to the natural course of the disease:

SARS-CoV2 infection: Positive PCR result for SARS-CoV2 or a clinical diagnosis made by a Primary Care physician, or a hospital discharge report stating a diagnosis of COVID-19 (ICD-10 codes used are displayed in Appendix A), from 25 February 2020 to 30 April 2020. Time (in days) from 24 February 2020 until a positive PCR or a clinical diagnosis (the first event) was used for survival analysis. Censored time for those individuals without the event was the time from 24 February to 30 April 2020.

COVID-19 mortality: Death in patients diagnosed with COVID-19 infection, between 25 February and 15 May. Patients with COVID-19 admitted to hospital before 16 May 2020 but resulting in death before 7 June were also included. Time (in days) from 24 February 2020 to COVID-19 death was used for survival analysis. Censored time for those individuals without the event was the time from 24 February to 7 June 2020.

Severe COVID-19: Composite outcome of COVID-19 mortality, as already defined, or COVID-19 hospital admission needing non-invasive mechanical ventilation, orotracheal intubation, mechanical ventilation or intensive care unit admission from 25 February 2020 to 15 May 2020. Time (in days) from 24 February 2020 until hospital admission (if severe COVID-19 developed during hospitalization) or time (in days) from 24 February 2020 until COVID-19 death was used for the survival analysis. Censored time for those individuals without the event was the time from 24 February to 7 June 2020.

### 2.6. Assessment of Additional Covariates

In addition to the covariates used for matching, chronic kidney disease (CKD) stages were identified based on eGFR [18]. Patients on renal replacement therapy (chronic hemodialysis or peritoneal dialysis) were identified and assigned to stage 5 CKD irrespective of their serum creatinine levels. Additional covariates not used for matching, but included in multivariate analyses, were the diagnosis of parathyroid disease (see ICD-10 codes used in Appendix A) and the status of renal transplant carrier. Once the patients without available serum creatinine levels were excluded, there were no missing values in other variables. 

### 2.7. Statistical Analysis

Continuous variables are reported as mean and standard deviation and qualitative variables are summarized by frequencies and percentages. Basal differences between treated and untreated groups were assessed using Student’s *t* test or chi-squared test and standardized mean differences. 

Once the control group for calcitriol was established, Kaplan–Meier curves were plotted for each outcome variable and Log-rank tests were performed to assess the differences between treated and not treated patients. Associations between calcitriol supplementation and outcome variables were further analyzed using unadjusted and multivariate Cox proportional hazards regression models. Finally, the association between the mean daily calcitriol dose and COVID-19 outcomes were also analyzed using multivariate Cox regression analysis.

For all statistical tests, a *p*-value < 0.05 was used for statistical significance. Descriptive statistics and survival analysis were carried out using SPSS version 25.0 for Windows (SPSS, Chicago, IL, USA) and Survival and Survminer packages in R [19,20].

### 2.8. Ethical Issues and Confidenciality

All data were treated anonymously in order for this study to comply with the provisions of Spanish and European laws on Protection of Personal Data. The study was approved by the ethics committee of the Parc Taulí Health Corporation Consortium—Autonomous University of Barcelona. 

## 3. Results

A total of 8076 patients ≥ 18 years-old were identified as being on calcitriol treatment in Catalonia (Spain) between 1 April 2019 and 28 February 2020. After propensity score matching, 6252 patients on calcitriol and 12,504 matched control patients were included in the study. 

The main clinical variables for patients treated with calcitriol and their matched controls are shown in Table 1; a detailed description of all matched and unmatched variables is offered in Appendix A. The balance of the matched covariates was considered satisfactory since all standardized mean differences were <0.10. Mean age of treated patients was 70.2 years and there was a slight female predominance. A high proportion of patients on calcitriol treatment and their matched controls were diagnosed with chronic kidney disease (CKD), 68% of cases in stages ≥3 CKD, with associated comorbidities such as hypertension, diabetes, ischemic heart disease or heart failure. Hypoparathyroidism was the second most frequent indication for calcitriol treatment, being diagnosed in 18% of the patients in the calcitriol-treated group.

### 3.1. Association between Calcitriol Supplementation and SARS-CoV2 Infection 

Among patients treated with calcitriol, 328 (5.2%) were diagnosed with SARS-CoV2 infection during the period of study, while 703 (5.6%) patients developed SARS-CoV2 infection in the control group.

The Kaplan–Meier plot did not show any significant reduction in the risk of SARS-CoV2 infection in patients supplemented with calcitriol (Figure 1a). In addition, neither univariate, nor multivariate Cox regression analysis showed any significant association between calcitriol supplementation and a reduced risk of SARS-CoV2 infection in the whole cohort. However, in patients in stages 4 or 5 of CKD, calcitriol treatment was associated with a significant reduction in the rate of SARS-CoV2 infection compared with untreated controls (163/2296 [7.1%] vs. 326/3407 [9.6%]; HR 0.78 [CI 95% 0.64–0.94]; *p* = 0.010) (Table 2 and Figure 1b). 

Mean daily calcitriol dose in 5885 subjects supplemented from November 2019 to February 2020 was 264.9 μg/day (SD 217.5). Neither univariate nor multivariate Cox regression analysis confirmed any association between the dose supplied and the risk of SARS-CoV2 infection in all treated patients (Table 3), nor in the subgroup of treated patients with advanced CKD (HR 0.98 [CI 95% 0.70–1.37]; *p* = 0.89). 

### 3.2. Association between Calcitriol Supplementation and Risk of Severe COVID-19 or COVID-19 Mortality

Among patients treated with calcitriol, 85 (1.4%) developed severe COVID-19 and 76 (1.2%) died due to COVID-19, while in the matched control group, 241 (1.9%) developed severe COVID-19 and 208 (1.7%) died due COVID-19.

Kaplan–Meier plots showed significant reductions in the risk of severe COVID-19 and COVID-19 mortality in patients supplemented with calcitriol (Figure 2). 

Univariate and multivariate Cox regression analyses also showed that calcitriol treatment was associated with significant lower risk of severe COVID-19 (HR 0.68 [CI 95% 0.53–0.87], *p* = 0.002) (Table 4) and lower risk of COVID-19 mortality (HR 0.66 [CI 95% 0.51–0.86], *p* = 0.002) (Table 5) in the whole cohort. In addition, among SARS-CoV2 infected patients (*n* = 1031), a significant lower mortality was observed in those supplemented with calcitriol (HR 0.75 [CI 95% 0.57–0.97]; *p* = 0.031). 

When performing subgroup analysis by CKD stages, important reductions in COVID-19 severity (HR 0.57 [CI 95% 0.41–0.79]; *p* = 0.001) and mortality (HR 0.57 [CI 95% 0.41–0.80]; *p* = 0.001) were observed in patients in stages 4 or 5 CKD (Table 4 and Table 5). Similar results were obtained if patients on renal replacement therapy were included or excluded from the analysis. Finally, no significant differences in severity or mortality with respect to the control group were found in patients in earlier stages of CKD. 

Among the 5885 patients that had been supplemented with calcitriol between 1 November 2019 and 28 February 2020, a progressive decline in the risk of severe COVID-19 or COVID-19 mortality was observed with increasing doses of calcitriol (Figure 3).

In the multivariate Cox regression analysis, the mean daily dose of calcitriol, measured in 0.25-μg intervals, was associated with significant reductions in the risk of severe COVID-19 (HR 0.45 [CI 95% 0.27–0.72], *p* = 0.001) or the risk of COVID-19 mortality (HR 0.53 [CI 95% 0.30–0.94], *p* = 0.030), independently of renal function or other comorbid conditions (Table 3). 

## 4. Discussion

To the best of our knowledge, this is the first study that analyzes the associations between calcitriol supplementation, the active metabolite of vitamin D and COVID-19 outcomes. 

Several clinical trials and two metanalysis have shown beneficial effects of cholecalciferol or ergocalciferol supplementation to prevent respiratory infections [21,22]. However, at present, it is unknown if vitamin D supplementation may exert any preventive or therapeutic effect on SARS-CoV2 infection. Two small-sized observational studies have shown divergent results, either with a trend to an increased mortality in patients supplemented with calcifediol [23] or a better survival in geriatric patients under cholecalciferol supplementation [24]. In addition, three low-powered clinical trials using cholecalciferol or calcifediol supplementation in hospitalized patients with COVID-19 have not observed any significant reduction in mortality [25,26,27].

In this large population-based cohort, we observed significant reductions in the risk of severe COVID-19 and COVID-19 mortality in patients supplemented with calcitriol compared to matched controls. These associations were remarkable in patients in stages 4 or 5 CKD, where calcitriol use was associated with 43% reduction in COVID-19 mortality. Patients with advanced CKD may have lower endogenous synthesis of calcitriol [28] due to impaired renal 1-hydroxylase activity, so that reduced mortality in this subgroup of patients could be the result of restoring the physiologic levels of the active hormone. However, we also found an inverse association between the calcitriol dose being supplied and the risk of severe COVID-19 and COVID-19 mortality, even in patients with normal renal function. This would suggest that supraphysiologic levels of calcitriol may also be of benefit in the defense against COVID-19. 

Our results should be viewed with caution. Although this study is population-based and the multivariate analysis indicates that calcitriol treatment is an independent variable associated with significantly better COVID-19 outcomes, the population on calcitriol treatment is specially enriched in patients diagnosed with CKD and associated comorbidities; thus, these results cannot be extrapolated to the general population. 

Calcitriol is often used to prevent or treat mineral and bone disorders associated with CKD. According to KDIGO, which offers evidence-based clinical practice guidelines for kidney disease, pre-COVID-19 instructions offer that calcitriol supplements should be reserved for patients with CKD stages 4 or 5 with severe and progressive secondary hyperparathyroidism [29]. Our results suggest that calcitriol might be used in patients with advanced kidney disease, especially during this pandemic, to reduce COVID-19-related mortality. 

We did not find lower rates of SARS-CoV2 infection with the use of calcitriol in the whole cohort, and the use of higher doses of calcitriol were not associated with lower risk of SARS-CoV2 infection in treated patients. These results would suggest that pathophysiological mechanisms that intervene in the process of infection are different from those that take place in the minority of patients that develop severe lung or systemic inflammation. Calcitriol exerts anti-inflammatory effects that can be mediated through several mechanisms, such as decreasing the production of pro-inflammatory cytokines [30], inhibiting the prostaglandin pathway [31], reducing the synthesis of angiotensin II and increasing the production of angiotensin (1–7) [11] or inhibiting bradykinin receptor expression [12]. It is tempting to speculate that some of these mechanisms could explain the lower severity and mortality of COVID-19 observed in our patients on calcitriol supplementation. 

We think that our study has some strengths, including the assessment of COVID-19 outcomes in a large population under calcitriol supplementation and the use of a matched cohort of controls. This study also has some limitations. First, there are the limitations of an observational cohort. Although we were comprehensive in analyzing many covariables, it is possible that there are still important covariables not considered in the matching process that may disbalance the treated and control groups. For this reason, we also analyzed the dosing of calcitriol to assess the associations between calcitriol doses and COVID-19 outcomes within the treated group, observing similar results. Second, our data were obtained from the registries of the health administration of the government of Catalonia, which are fed by the diagnoses issued by family physicians, hospital discharge reports or medicines supplied by pharmacies, with the inherent limitations of administrative data. Third, our population is mainly of Caucasian origin and we cannot exclude that our results may be different in other ethnic groups. Finally, we decided to focus our analysis on the first wave of the pandemic, with a higher number of severe cases and mortality. However, the diagnosis of SARS-CoV2 in that phase could not be ascertained with PCR in all the cases, and some patients received a clinical diagnosis without a confirmatory microbiological confirmation.

## 5. Conclusions

In this large, population-based study, we have shown that supplementation with calcitriol was associated with significant reductions in COVID-19 severity and mortality, particularly in patients with advanced CKD. In our opinion, a clinical trial to confirm the effects of calcitriol on COVID-19 would be justified. Meanwhile, calcitriol supplementation should be considered in patients with CKD during the COVID-19 pandemic.

## Figures and Tables

**Figure 1 biomedicines-09-00509-f001:**
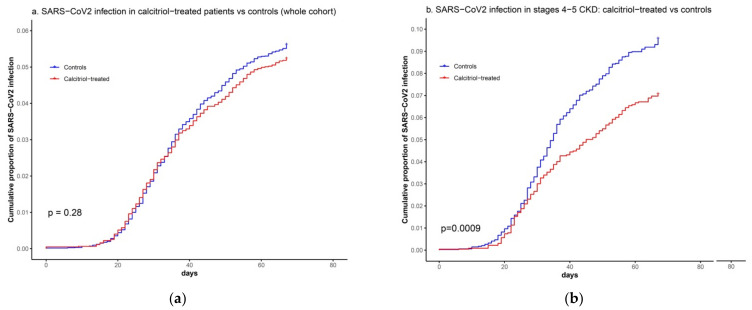
Kaplan–Meier plots showing (**a**) the cumulative proportion of patients with SARS-CoV2 infection between 25 February 2020 and 30 April 2020 in calcitriol-treated patients and matched controls (whole cohort); and (**b**) the cumulative proportion of SARS-CoV2 infection in patients in stages 4 or 5 CKD treated with calcitriol versus untreated controls.

**Figure 2 biomedicines-09-00509-f002:**
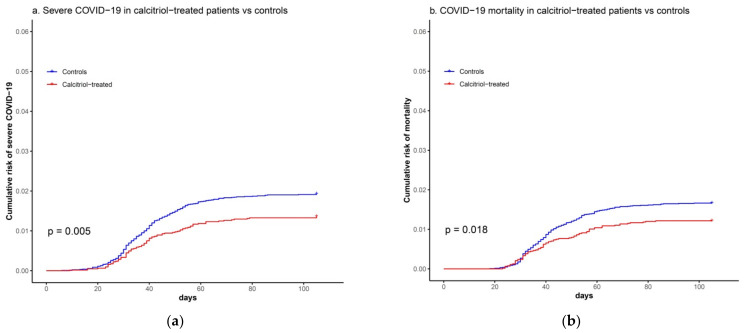
Kaplan–Meier plots showing: (**a**) Cumulative risk of severe COVID-19 (composite outcome of need for non-invasive mechanical ventilation, orotracheal intubation, mechanical ventilation, intensive care unit admission or COVID-19 mortality) between 25 February 2020 and 7 June 2020 in patients supplemented with calcitriol or matched controls or (**b**) COVID-19 mortality in patients supplemented with calcitriol or matched controls during the same period.

**Figure 3 biomedicines-09-00509-f003:**
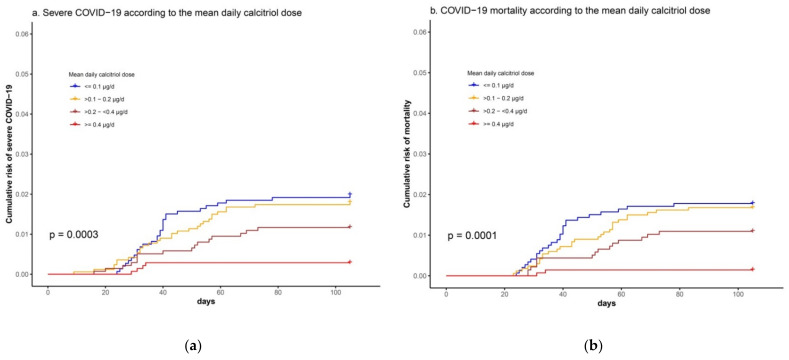
Kaplan–Meier plots showing (**a**) cumulative risk of severe COVID-19 (composite outcome of need for non-invasive mechanical ventilation, orotracheal intubation, mechanical ventilation, intensive care unit admission or COVID-19 mortality) between 25 February 2020 and 7 June 2020, according to the calcitriol dose being supplied (by quartiles); (**b**) COVID-19 mortality during the same period, according to the calcitriol dose being supplied (by quartiles).

**Table 1 biomedicines-09-00509-t001:** Main clinical characteristics of the patients on calcitriol treatment and matched controls.

Variables	Calcitriol-Treated (*n* = 6252)	Matched Controls (*n* = 12,504)	P ^1^	SMD ^2^
Female gender, *n*(%)	3596 (57.5)	7185 (57.5)	0.954	0.001
Age, mean (SD)	70.2 (15.6)	70.7 (14.7)	0.022	0.035
Cigarette smoking, *n*(%)	1762 (28.2)	3562 (28.5)	0.676	0.007
Nursing home residence, *n*(%)	136 (2.2)	270 (2.2)	0.986	0.001
Hypertension, *n*(%)	4308 (68.9)	8686 (69.5)	0.443	0.012
Obesity, *n*(%)	2637 (42.2)	5430 (43.4)	0.107	0.025
Diabetes, *n*(%)	2355 (37.7)	4857 (38.8)	0.123	0.024
Heart failure, *n*(%)	1691 (27.0)	3633 (29.1)	0.004	0.045
COPD, *n*(%)	1255 (20.1)	2638 (21.1)	0.107	0.025
Asthma, *n*(%)	581 (9.3)	1228 (9.8)	0.259	0.018
eGFR, mean (SD)	49.0 (30.8)	48.7 (26.6)	0.379	0.013
Cerebrovascular disease, *n*(%)	696 (11.1)	1487 (11.9)	0.132	0.024
Dementia, *n*(%)	345 (5.5)	744 (6.0)	0.246	0.019
Malignant neoplasia, *n*(%)	2451 (39.2)	5215 (41.7)	0.001	0.051
Liver cirrhosis, *n*(%)	106 (1.7)	235 (1.9)	0.406	0.014

^1^ chi-squared (dichotomous variables) or Student’s *t* test (continuous variables). ^2^ Standardized mean difference. COPD: chronic obstructive pulmonary disease. eGFR: estimated glomerular filtration rate. SD: Standard deviation.

**Table 2 biomedicines-09-00509-t002:** Variables associated with SARS-CoV2 infection ^1^ in patients on calcitriol treatment and matched controls.

Variables	Overall Cohort ^2^		Subjects Diagnosed with CKD, Stages 4–5 ^3^	
	Univariate ^4^ HR (CI 95%)p ^6^	Multivariate ^5^ HR (CI 95%)p ^6^	Univariate ^4^ HR (CI 95%)p ^6^	Multivariate ^5^ HR (CI 95%)p ^6^
Calcitriol treatment	0.93 (0.82–1.06)		0.73 (0.60–0.88) ***	0.78 (0.64–0.94) **
Female sex	0.85 (0.75–0.96) **		0.87 (0.72–1.03)	0.78 (0.65–0.93) **
Age ^7^	1.13 (1.08–1.18) ***		1.07 (0.99–1.16)	
Cigarette smoking	1.12 (0.99–1.28)		1.11 (0.92–1.34)	
Nursing home residence	6.06 (4.99–7.37) ***	4.23 (3.42–5.22) ***	4.55 (3.50–5.92) ***	4.03(3.03–5.34) ***
Hypertension	1.21 (1.05–1.39) **		0.89 (0.70–1.14)	
Obesity	1.11 (0.98–1.25)		1.24 (1.04–1.48) *	1.37 (1.14–1.64) ***
Diabetes	1.30 (1.15–1.47) ***		1.10 (0.92–1.32)	
Heart failure	1.75 (1.54–1.98) ***	1.24 (1.08–1.42) **	1.36 (1.14–1.62) ***	1.21 (1.01–1.45) *
COPD	1.48 (1.29–1.69) ***	1.21 (1.05–1.39) **	1.19 (0.98–1.44)	
Asthma	1.06 (0.86–1.29)		0.97 (0.71–1.32)	
eGFR ^8^	0.88 (0.86–0.90) ***	0.93 (0.90–0.96) ***	0.74 (0.66–0.83) ***	0.74 (0.66–0.83) ***
Cerebrovascular disease	1.53 (1.30–1.81) ***	1.20 (1.02–1.42) *	1.13 (0.89–1.42)	
Dementia	2.54 (2.12–3.04) ***	1.64 (1.35–1.99) ***	2.16 (1.69–2.75) ***	1.66 (1.28–2.16) ***
Malignant neoplasia	1.17 (1.03–1.32) *		1.17 (0.98–1.40)	
Liver cirrhosis	1.02 (0.65–1.61)		1.28 (0.78–2.11)	
Osteoporosis	0.97 (0.79–1.20)		0.88 (0.64–1.22)	
Dyslipidemia	1.05 (0.93–1.19)		1.12 (0.93–1.34)	
Ischemic heart disease	1.18 (1.01–1.38) *		0.97 (0.79–1.20)	
Peripheral arteriopathy	1.46 (1.20–1.77) ***		1.17 (0.91–1.49)	
Hypoparathyroidism	0.55 (0.40–0.77) ***		0.22 (0.09–0.54) ***	
Use of PPI	1.43 (1.26–1.63) ***		1.22 (1.00–1.50) *	
Use oral corticosteroids	1.35 (1.15–1.56) ***	1.25 (1.06–1.46) **	1.15 (0.91–1.44)	
Use of DPP4-inhibitors	1.22 (1.02–1.46) *		0.83 (0.66–1.06)	
Use of statins	0.94 (0.83–1,06)	0.81 (0.71–0.92) ***	0.75 (0.63–0.89) ***	0.78 (0.65–0.94) **
Use of ACE inhibitors	0.84 (0.72–0.99) *		0.85 (0.66–1.09)	
Use of ARB	0.89 (0.77–1.03)		0.76 (0.62–0.95) *	
Use of immunosuppressants	1.03 (0.84–1.28)		1.24 (0.92–1.68)	1.53 (1.13–2.07) **
Renal replacement therapy	2.38 (1.97–2.86) ***	1.68 (1.36–2.06) ***	1.53 (1.25–1.87) ***	
Kidney transplant carrier	0.91 (0.68–1.22)		1.07 (0.71–1.61)	

^1^ Positive PCR or clinical diagnosis of SARS-CoV2 infection. ^2^ Patients on calcitriol treatment (*n* = 6252) and controls (*n* = 12,504). ^3^ Patients diagnosed with chronic kidney disease stages 4 or 5, on calcitriol treatment (*n* = 2296) or untreated controls (*n* = 3407). ^4^ Unadjusted Cox regression analysis. ^5^ Cox regression analysis controlling for all covariates. ^6^ **p* ≤ 0.05; ** *p* ≤ 0.01; ****p* ≤ 0.001. ^7^ Ratios are calculated for every 10 years of age. ^8^ eGFR: estimated glomerular filtration rate. Ratios are calculated for every 10-mL increase of creatinine clearance. HR: hazard ratio. CI 95%: confidence interval 95%. COPD: chronic obstructive pulmonary disease. PPI: proton pump inhibitors. DPP4: dipeptidyl peptidase-4. ACE: angiotensin-converting enzyme. ARB: angiotensin-II receptor blockers.

**Table 3 biomedicines-09-00509-t003:** Associations between mean daily calcitriol dose and COVID-19 outcomes (*n* = 5885).

Variables	SARS-CoV2 Infection ^1^		Severe COVID-19 ^2^		COVID-19 Mortality	
	Univariate ^3^ HR (CI 95%)p ^5^	Multivariate ^4^ HR (CI 95%)p ^5^	Univariate ^3^ HR (CI 95%)p ^5^	Multivariate ^4^ HR (CI 95%)p ^5^	Univariate ^3^ HR (CI 95%)p ^5^	Multivariate ^4^ HR (CI 95%)p ^5^
Calcitriol dose	0.90 (0.78–1.03)		0.40 (0.25–0.62) ***	0.45 (0.27–0.72) ***	0.35 (0.21–0.58) ***	0.53 (0.30–0.94) *
Female sex	1.03 (0.82–1.29)		0.78 (0.50–1.21)		0.87 (0.55–1.39)	
Age ^6^	1.12 (1.04–1.21) **		1.84 (1.49–2.26) ***	1.38 (1.11–1.72) **	2.17 (1.71–2.77) ***	1.42 (1.07–1.87) *
Cigarette smoking	1.16 (0.91–1.47)		1.06 (0.65–1.72)		1.15 (0.70–1.91)	
Nursing home residence	8.51 (6.17–11.76) ***	8.02 (5.78–11.12)***	12.42 (7.17–21.50) ***	6.03 (3.30–11.01) ***	14.31 (8.20–24.98) ***	6.45 (3.51–11.85) ***
Hypertension	1.09 (0.85–1.39)		1.92 (1.09–3-36) *		2.01 (1.10–3.66) *	
Obesity	1.13 (0.90–1.42)		1.47 (0.95–2.29)		1.66 (1.04–2.65) *	
Diabetes	1.12 (0.89–1.41)		1.53 (0.98–2.37)		1.60 (1.01–2.55) *	
Heart failure	1.75 (1.39–2.21) ***	1.50 (1.18–1.90)***	3.19 (2.05–4.97) ***	1.66 (1.04–2.65) *	3.65 (2.28–5.85) ***	1.79 (1.09–2.94) *
COPD	1.30 (1.00–1.69) *		1.27 (0.76–2.13)		1.36 (0.80–2.33)	
Asthma	1.14 (0.79–1.64)		0.79 (0.34–1.82)		0.89 (0.39–2.05)	
eGFR ^7^	0.94 (0.91–0.98) **		0.77 (0.69–0.86) ***		0.68 (0.60–0.78) ***	0.79 (0.67–0.94) **
Cerebrovascular disease	1.43 (1.04–1.95) *		1.85 (1.06–3.25) *		1.95 (1.08–3.49) *	
Dementia	2.33 (1.64–3.31) ***		4.91 (2.87–8.39) ***	2.32 (1.30–4.16) **	5.64 (3.27–9.72) ***	2.37 (1.32–4.25) **
Malignant neoplasia	1.06 (0.84–1.34)		1.47 (0.94–2.28)	1.57 (1.00–2.44) *	1.63 (1.02–2.59) *	1.72 (1.07–2.78) *
Liver cirrhosis	0.90 (0.37–2.18)		0.05 (0–75.08)		0.05 (0–112.61)	
Osteoporosis	0.90 (0.60–1.34)		0.94 (0.43–2.04		1.06 (0.48–2.30)	
Dyslipidemia	0.92 (0.74–1.16)		1.64 (1.04–2.60) *		1.76 (1.08–2.86) *	
Ischemic heart disease	1.00 (0.74–1.37)		1.05 (0.58–1.91)		1.09 (0.58–2.02)	
Peripheral arteriopathy	1.27 (0.87–1.85)		2.07 (1.12–3.83) *		1.90 (0.97–3.71	
Hypoparathyroidism	0.62 (0.43–0.88) **		0.19 (0.06–0.60)		0.07 (0.10–0.49) **	
Use of PPI	1.27 (1.01–1.60) *		1.76 (1.09–2.85) *		1.83 (1.10–3.05) *	
Use oral corticosteroids	1.53 (1.16–2.01) **		2.24 (1.38–3.64) ***	2.28 (1.39–3.73) ***	1.99 (1.18–3.36) **	
Use of DPP4-inhibitors	1.02 (0.71–1.46)		1.50 (0.81–2.77)		1.53 (0.81–2.92)	
Use of statins	0.89 (0.71–1.12)		1.27 (0.81–1.97)		1.15 (0.72–1.84)	
Use of ACE inhibitors	0.92 (0.69–1.23)		0.84 (0.47–1.50)		0.72 (0.38–1.36)	
Use of ARB	0.81 (0.61–1.06)		0.99 (0.59–1.64)		0.92 (0.53–1.58)	
Use of immunosuppressants	1.25 (0.89–1.76)		1.56 (0.84–2.88)		1.26 (0.63–2.53)	
Renal replacement therapy	4.14 (2.84–6.04) ***	3.83(2.61–5.61) ***	1.88 (0.69–5.16)		2.12 (0.77–5.82)	
Kidney transplant carrier	1.23 (0.77–1.96)		1.44 (0.63–3.33)		1.06 (0.38–2.91)	

^1^ Positive PCR or clinical diagnosis of SARS-CoV2 infection. ^2^ Composite outcome of need for non-invasive mechanical ventilation, orotracheal intubation, mechanical ventilation, intensive care unit admission or death. ^3^ Unadjusted Cox regression analysis. ^4^ Cox regression analysis controlling for all covariates. ^5^ * *p* ≤ 0.05; ** *p* ≤ 0.01; *** *p* ≤ 0.001. ^6^ Ratios are calculated for every 10 years of age. ^7^ eGFR: estimated glomerular filtration rate. Ratios are calculated for every 10-mL increase of creatinine clearance. HR: hazard ratio. CI 95%: confidence interval 95%. COPD: chronic obstructive pulmonary disease. PPI: proton pump inhibitors. DPP4: dipeptidyl peptidase-4. ACE: angiotensin-converting enzyme. ARB: angiotensin-II receptor blockers.

**Table 4 biomedicines-09-00509-t004:** Variables associated with severe COVID-19^1^ in patients on calcitriol treatment and matched controls.

Variables	Overall Cohort ^2^		Subjects Diagnosed with CKD, Stages 4–5 ^3^	
	**Univariate ^4^** **HR (CI 95%)p ^6^**	**Multivariate ^5^** **HR (CI 95%)p ^6^**	**Univariate ^4^** **HR (CI 95%)p ^6^**	**Multivariate ^5^** **HR (CI 95%)p ^6^**
Calcitriol treatment	0.70 (0.55–0.90) **	0.68 (0.53–0.87) **	0.51 (0.37–0.70) ***	0.57 (0.41–0.79) ***
Female sex	0.54 (0.43–0.67) ***	0.68 (0.54–0.86) ***	0.82 (0.61–1.08)	
Age ^7^	1.57 (1.43–1.73) ***	1.18 (1.06–1.32) **	1.19 (1.04–1.35) ***	
Cigarette smoking	1.12 (0.88–1.42)		1.09 (0.81–1.47)	
Nursing home residence	6.86 (4.96–9.48) ***	3.58 (2.52–5.08) ***	5.01 (3.38–7.43) ***	3.64 (2.39–5.55) ***
Hypertension	1.82 (1.39–2.40) ***		0.81 (0.55–1.17)	
Obesity	1.19 (0.96–1.48)		1.21(0.91–1.60)	
Diabetes	1.73 (1.39–2.15) ***		1.02 (0.77–1.35)	
Heart failure	2.74 (2.21–3.41) ***	1.44 (1.15–1.81) **	1.61 (1.21–2.13) ***	1.45 (1.09–1.93) **
COPD	1.82 (1.45–2.30) ***		1.24 (0.92–1.66)	
Asthma	1.02 (0.71–1.47)		1.20 (0.77–1.89)	
eGFR ^8^	0.72 (0.68–0.76) ***	0.77 (0.73–0.82) ***	0.76 (0.63–0.91) **	0.77 (0.64–0.93) **
Cerebrovascular disease	1.83 (1.39–2.41) ***		1.14 (0.79–1.64)	
Dementia	4.19 (3.19–5.48) ***	2.37 (1.76–3.10) ***	3.04 (2.16–4.27) ***	2.33 (1.61–3.37) ***
Malignant neoplasia	1.54 (1.24–1.92) ***	1.26 (1.00–1-58) *	1.42 (1.08–1.88) **	1.41 (1.07–1.87) *
Liver cirrhosis	1.02 (0.45–2.29)		1.19 (0.53–2.68)	
Osteoporosis	0.91 (0.62–1.34)		0.88 (0.53–1.47)	
Dyslipidemia	1.32 (1.06–1.64) *		1.36 (1.01–1.82) *	1.51 (1.11–2.04) **
Ischemic heart disease	1.61 (1.25–2.08) ***		1.00 (0.72–1.39)	
Peripheral arteriopathy	2.16 (1.61–2.91) ***		1.42 (0.99–2.03)	
Hypoparathyroidism	0.19 (0.07–0.50) ***		0.12 (0.02–0.87) *	
Use of PPI	2.04 (1.60–2.61) ***		1.22 (0.89–1.67)	
Use oral corticosteroids	1.41 (1.07–1.86) **		1.12 (0.78–1.60)	
Use of DPP4-inhibitors	1.16 (0.93–1.44)		0.67 (0.45–1.01)	
Use of statins	1.37 (1.01–1.87) *		0.74 (0.56–0.98) *	0.74 (0.55–0.99) *
Use of ACE inhibitors	0.86 (0.65–1.14)		0.79 (0.52–1.19)	
Use of ARB	0.88 (0.68–1.14)		0.68 (0.48–0.97) *	
Use of immunosuppressants	1.36 (0.97–1.90)	1.65 (1.17–2.33) **	1.47 (0.94–2.29)	1.83 (1.17–2.86) **
Renal replacement therapy	1.97 (1.39–2.79) ***		0.91 (0.63–1.32)	
Kidney transplant carrier	1.25 (0.80–1.97)		1.22 (0.66–2.24)	

^1^ Composite outcome of the need for non-invasive mechanical ventilation, orotracheal intubation, mechanical ventilation, intensive care unit admission or death due to COVID-19. ^2^ Patients on calcitriol treatment (*n* = 6252) and controls (*n* = 12,504). ^3^ Patients diagnosed with chronic kidney disease stages 4 or 5, on calcitriol treatment (*n* = 2296) or untreated controls (*n* = 3407). ^4^ Unadjusted Cox regression analysis. ^5^ Cox regression analysis controlling for all covariates. ^6^ * *p* ≤ 0.05; ** *p* ≤ 0.01; *** *p* ≤ 0.001. ^7^ Ratios are calculated for every 10 years of age. ^8^ eGFR: estimated glomerular filtration rate. Ratios are calculated for every 10-mL increase of creatinine clearance. HR: hazard ratio. CI 95%: confidence interval 95%. COPD: chronic obstructive pulmonary disease. PPI: proton pump inhibitors. DPP4: dipeptidyl peptidase-4. ACE: angiotensin-converting enzyme. ARB: angiotensin-II receptor blockers.

**Table 5 biomedicines-09-00509-t005:** Variables associated with COVID-19 mortality in patients on calcitriol treatment and matched controls.

Variables	Overall Cohort ^1^		Subjects Diagnosed with CKD, Stages 4–5 ^2^	
	**Univariate ^3^** **HR (CI 95%)p ^5^**	**Multivariate ^4^** **HR (CI 95%)p ^5^**	**Univariate ^3^** **HR (CI 95%)p ^5^**	**Multivariate ^4^** **HR (CI 95%)p ^5^**
Calcitriol treatment	0.73 (0.56–0.95) *	0.66 (0.51–0.86) **	0.55 (0.39–0.76) ***	0.57 (0.41–0.80) ***
Female sex	0.56 (0.45–0.71 )***	0.70 (0.54–0.89) **	0.83 (0.61–1.11)	
Age ^6^	1.92 (1.71–2.15) ***	1.39 (1.22–1.58) ***	1.42 (1.22–1.65) ***	1.42 (1.20–1.67) ***
Cigarette smoking	1.15 (0.90–1.48)		1.13 (0.83–1.55)	1.44 (1.04–2.00) *
Nursing home residence	8.08 (5.82–11.21) ***	3.63 (2.54–5.18) ***	5.72 (3.84–8.51) ***	4.16 (2.70–6.42) ***
Hypertension	2.08 (1.54–2.83) ***		0.79 (0.54–1.18)	
Obesity	1.26 (0.99–1.58)		1.24 (0.93–1.67)	1.47 (1.09–1.99) *
Diabetes	1.88 (1.49–2.37) ***		1.06 (0.79–1.43)	
Heart failure	3.24 (2.57–4.10) ***	1.55 (1.21–1.98) ***	1.76 (1.30–2.38) ***	
COPD	2.02 (1.59–2.59) ***		1.36 (1.00–1.85) *	
Asthma	1.11 (0.76–1.62)		1.21 (0.75–1.95)	
eGFR ^7^	0.69 (0.65–0.73) ***	0.72 (0.66–0.78) ***	0.78 (0.64–0.94) **	0.73 (0.60–0.89) **
Cerebrovascular disease	1.87 (1.40–2.51) ***		1.12 (0.76–1.65)	
Dementia	4.91 (3.72–6.47) ***	2.37 (1.75–3.21) ***	3.39 (2.39–4.81) ***	2.35 (1.60–3.43) ***
Malignant neoplasia	1.67 (1.32–2.11) ***	1.34 (1.05–1.72) *	1.57 (1.17–2.11) **	1.48 (1.10–2.00) **
Liver cirrhosis	0.58 (0.18–1.79)		0.64 (0.20–2.00)	
Osteoporosis	0.93 (0.62–1.41)		0.79 (0.45–1.39)	
Dyslipidemia	1.38 (1.09–1.75) **		1.39 (1.02–1.90) *	
Ischemic heart disease	1.67 (1.28–2.20) ***		1.02 (0.73–1.44)	
Peripheral arteriopathy	2.14 (1.56–2.95) ***		1.29 (0.87–1.91)	
Hypoparathyroidism	0.11 (0.03–0.43) **		0.14 (0.02–0.98) *	
Use of PPI	2.21 (1.70–2.88) ***		1.36 (0.97–1.92)	
Use oral corticosteroids	1.38 (1.03–1.86) *		1.10 (0.75–1.62)	
Use of DPP4-inhibitors	1.46 (1.06–2.01) *		0.70 (0.46–1.07)	
Use of statins	1.11 (0.88–1.41)		0.70 (0.52–0.94) *	
Use of ACE inhibitors	0.81 (0.59–1.10)		0.73 (0.47–1.14)	
Use of ARB	0.9 1 (0.69–1.19)		0.70 (0.50–0.98) *	
Use of immunosuppressants	1.16 (0.79–1.71)	1.60 (1.08–2.37) *	1.23 (0.75–2.04)	1.96 (1.17–3.28) **
Renal replacement therapy	1.78 (1.21–2.63) **	0.64 (0.42–0.99) *	0.79 (0.53–1.19)	
Kidney transplant carrier	0.92 (0.53–1.60)		0.84 (0.40–1.80)	

^1^ Patients on calcitriol treatment (*n* = 6252) and controls (*n* = 12,504). ^2^ Patients diagnosed with chronic kidney disease stages 4 or 5, on calcitriol treatment (*n* = 2296) or untreated controls (*n* = 3407). ^3^ Unadjusted Cox regression analysis. ^4^ Cox regression analysis controlling for all covariates. ^5^ * *p* ≤ 0.05; ** *p p* ≤ 0.01; *** *p* ≤ 0.001. ^6^ Ratios are calculated for every 10 years of age. ^7^ eGFR: estimated glomerular filtration rate. Ratios are calculated for every 10-mL increase of creatinine clearance. HR: hazard ratio. CI 95%: confidence interval 95%. COPD: chronic obstructive pulmonary disease. PPI: proton pump inhibitors. DPP4: dipeptidyl peptidase-4. ACE: angiotensin-converting enzyme. ARB: angiotensin-II receptor blockers.

## Data Availability

All data can be supplied by the researchers upon request.

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
