# Peer review of "Association of Calcitriol Supplementation with Reduced COVID-19 Mortality in Patients with Chronic Kidney Disease: A Population-Based Study"

_biomedicines, 2021, doi:10.3390/biomedicines9050509_

Round 1

Reviewer 1 Report

This study aims to analyze the associations between calcitriol supplementation and the risk of SARS-CoV2 infection or COVID-19 mortality. The reviewer considers that this study to have a very large sample size and appropriate statistical analysis. However, there are some questions in this study.

This study examines the association between calcitriol supplementation and COVID-19 severity and mortality in CKD patients. Is the purpose of this study to compare the severity and mortality of COVID-19 with calcitriol supplementation in patients with and without CKD? Or is the purpose of this study to examine the difference in COVID-19 severity and mortality from calcitriol supplementation depending on advanced CKD? It is not clear why the authors conducted this study focusing on CKD, the background of this study, the process that led to the idea of this study, and the purpose of the study. Please clarify in the "Introduction" session why the authors conducted this study focusing on CKD, the background of this study, the process that led to the idea of this study, and the purpose of the study.

In relation to the above, the reviewer consider that it is also necessary to make the purpose of the study consistent with the conclusion.

This study compares calcitriol supplementation by the CKD stage with COVID-19 severity and mortality. The authors have specially sub-analyzed CKD stage 4-5, but how many patients on dialysis were included in the CKD stage 4-5 patients? The reviewer considers to need to examine the differences in calcitriol supplementation and COVID-19 severity and mortality with and without dialysis treatment.

The target population of this study is the inhabitants of Catalonia. Can the results of this study be applied to other ethnic groups?

Reviewer 2 Report

Very well pasper

Author Response

We appreciate the comment of the reviewer.

The reviewer points out that the Methods section could be improved. However, we have no more details on which specific section(s) should be modified.

Reviewer 3 Report

In the manuscript entitled: " Association of Calcitriol Supplementation with Reduced 2 COVID-19 Mortality in Patients with Chronic Kidney Disease: 3

a Population-based Study.", the authors analyze the associations between calcitriol supplementation and the risk of SARS-CoV2 infection or COVID-19 mortality. They presented evidences show that SARS-CoV2 infection rates, risk of severe COVID-19 and COVID-19 29 mortality is decreased in patient with calcitriol supplementation, but only in patients with CKD stage 4&5. The results are interesting, but there are some weakness and suggestion authors are asked to consider.

  1. Since calcitriol might reduce incidence of adult respiratory distress syndrome, is that possible that patients who take calcitriol have minor symptom and did not get SARS-COV2 test?
  2. Although authors address the anti-inflammatory effect of calcitriol, however, it can not explain why this protected effects only happened on CKD patients.

Author Response

Comments and Suggestions for Authors

In the manuscript entitled: " Association of Calcitriol Supplementation with Reduced COVID-19 Mortality in Patients with Chronic Kidney Disease:

a Population-based Study.", the authors analyze the associations between calcitriol supplementation and the risk of SARS-CoV2 infection or COVID-19 mortality. They presented evidences show that SARS-CoV2 infection rates, risk of severe COVID-19 and COVID-19 29 mortality is decreased in patient with calcitriol supplementation, but only in patients with CKD stage 4&5. The results are interesting, but there are some weakness and suggestion authors are asked to consider.

  1. Since calcitriol might reduce incidence of adult respiratory distress syndrome, is that possible that patients who take calcitriol have minor symptom and did not get SARS-COV2 test?

Response:

We did not find a significant reduction in the rate of SARS-CoV2 infection diagnoses in the whole cohort of patients treated with calcitriol as shown in Table 2. In our opinion, the fact that the number of infected patients was similar between treated patients and controls suggests that COVID-19 underdiagnosis in the calcitriol-treated group is unlikely. 

  1. Although authors address the anti-inflammatory effect of calcitriol, however, it can not explain why this protected effects only happened on CKD patients.

Response:

We found significant lower rates of severe COVID-19 and lower COVID-19 mortality in CKD patients supplemented with calcitriol. Patients with advanced CKD may have lower endogenous synthesis of calcitriol due to impaired renal 1-hydroxylation, so that reduced mortality in this subgroup of patients could be the result of restoring the physiological levels of the active hormone.

However, we also found a dose-effect of calcitriol that was independent of renal function, as stated in the last paragraph of the results section.

In the multivariate Cox regression analysis, the mean daily dose of calcitriol, measured in 0.25 μg intervals, was associated with significant reductions in the risk of severe COVID-19 (HR 0.45 [CI 95% 0.27-0.72], p=0.001) or the risk of COVID-19 mortality (HR 0.53 [CI 95% 0.30-0.94], p=0.030), independently of renal function or other comorbid conditions (Table 3). 

This would suggest that supraphysiological levels of calcitriol may probably be of benefit in the defense against COVID-19.   

We have introduced some modifications in the third paragraph of the Discussion:

"In this large population-based cohort, we observed significant reductions in the risk of severe COVID-19 and COVID-19 mortality in patients supplemented with calcitriol compared to matched controls. These associations were remarkable in patients in stages 4 or 5 CKD, where calcitriol use was associated with 43% reduction in COVID-19 mortality. Patients with advanced CKD may have lower endogenous synthesis of calcitriol [31] due to impaired renal 1-hydroxylase activity, so that reduced mortality in this subgroup of patients could be the result of restoring the physiological serum levels of the active hormone. However, we also found an inverse association between the calcitriol dose being supplied and the risk of severe COVID-19 and COVID-19 mortality, even in patients with normal renal function. This would suggest that supraphysiological levels of calcitriol may also be of benefit in the defense against COVID-19."   

Round 2

Reviewer 1 Report

I think all responses or comments to our questions have been addressed satisfactorily.

I have no comments on the revised manuscript.